# Relationship between clinical parameters and malformations in dogs diagnosed with atlanto-axial instability

**Harumichi Itoh**[1], **Takuya Itamoto**[1], **Kenji Tani**[2], **Hiroshi Sunahara**[2], **Yuki Nemoto**[2], **Munekazu Nakaichi**[3], **Toshie Iseri**[3], **Hiro Horikirizono**[3], **Kazuhito Itamoto**[1]*

1 Department of Small Animal Clinical Science, Joint Faculty of Veterinary Medicine, Yamaguchi University, Yamaguchi City, Yamaguchi, Japan, 2 Department of Veterinary Surgery, Joint Faculty of Veterinary Medicine, Yamaguchi University, Yamaguchi City, Yamaguchi, Japan, 3 Laboratory of Veterinary Radiology Yamaguchi University, Yamaguchi City, Yamaguchi, Japan

* kaz2356@yamaguchi-u.ac.jp

**Data Availability Statement:** All relevant data are within the manuscript and its Supporting Information files.

## Abstract

Atlanto-axial instability is a common disease that affects toy-breed dogs. Most cases of atlanto-axial instability are congenital. Furthermore, patients with atlanto-axial instability are predisposed to other concurrent diseases. Therefore, this study aimed to retrospectively determine the presence of concurrent diseases in cases with atlanto-axial instability using imaging data and analyze the relationship between clinical parameters and the incidence of complex malformations. The clinical data and imaging findings of 41 toy-breed dogs diagnosed with atlanto-axial instability were analyzed using their medical records and imaging data. Occipital dysplasia (17/27), atlanto-occipital overlapping (22/34), dens dysplasia (27/41), Chiari-like malformation (8/34), syringomyelia (5/34), lateral ventricular enlargement (20/36), and intracranial arachnoid cyst (5/35) were observed in patients with atlanto-axial instability. The body weight of the patients in the groups with atlanto-occipital overlapping and lateral ventricular enlargement was lower than that of those in the groups without these diseases (1.78 ± 0.71 vs 2.71 ± 1.15 kg, P = 0.0269, 1.60 ± 0.40 vs 2.75 ± 1.08 kg, P = 0.001, respectively). Furthermore, when the correlation between the total number of concurrent diseases and the age at onset and body weight was examined, it became clear that lower body weight was associated with the incidence of a greater number of concurrent diseases. Thus, the findings of this study suggest that toy-breed dogs are more likely to present with complex malformations and should be evaluated carefully with additional examinations and treatment methods.

## Introduction

The first case of atlanto-axial Instability (AAI) in dogs was reported by Geary [1]. Young toy-breed dogs, such as Yorkshire terriers, Pomeranians, Miniature/Toy poodles, Chihuahuas, Papillons, Maltese, and Miniature Dachshunds, are reportedly predisposed to congenital AAI [2–4]. The factors influencing onset include aplasia and hypoplasia of the dens due to

**Funding:** The author(s) received no specific funding for this work.

**Competing interests:** The authors have declared that no competing interests exist.

aberrations in physeal growth plate closure and abnormalities of the transverse ligament [3, 5]. The clinical symptoms of AAI include neck pain and progressive paraplegia. It has been suggested that AAI is caused by the compression of the spinal cord by dens due to the displacement of the axis. Minor trauma to congenitally abnormal joints may lead to the progression of clinical symptoms [6, 7].

The definitive diagnosis of AAI is made via radiographic examinations, computed tomography (CT), or magnetic resonance imaging (MRI) of the neck. CT scans provide useful information for the evaluation of dens dysplasia (DD) and other vertebral fractures, as well as treatment planning [8, 9]. Similarly, MRI also provides important information regarding the spinal cord, such as the presence of edema, hemorrhage, and syringomyelia [9]. In recent years, the presence of many concurrent diseases has been observed during CT and MRI examinations of patients with AAI. Some of these concurrent diseases are recognized as craniocervical junction abnormalities (CJA) [10]. CJA is a syndrome characterized by congenital dysplasia affecting the region extending from the occipital bone to the upper cervical spine. CJA includes conditions such as Chiari-like malformation (CLM), atlanto-occipital instability, AAI, occipital-atlas-axial dysplasia, atlanto-occipital overlapping (AOO), dorsal compression, caudal cerebral compression, and DD; however, the detailed pathogenic mechanism and diseases included in CJA are not clearly defined. Nevertheless, with the widespread use of CT and MRI scanning, an increasing number of abnormal imaging findings are being observed in patients with AAI.

Previous reports have shown significant differences in body weight in the group with AOO compared with that of the group without AOO [11]. However, since the analysis in this previous study focused only on AOO, the relationship between other concurrent diseases and body weight is unknown.

The purpose of this study was to retrospectively examine the imaging findings of toy-breed dogs diagnosed with AAI and analyze the correlation between the imaging findings and clinical data with regard to the incidence of concurrent diseases, including CJA.

## Materials and methods

### Ethics statement of clinical cases

Because this clinical study using owner's dogs was a retrospective study using imaging findings in clinical cases and because there was no apparent disadvantage to the animals, approval by the Animal Research Ethics Committee was not required. Additionally, all owners signed informed consent for the study using in-hospital imaging and other findings.

### Case selection

We retrospectively analyzed dogs diagnosed with AAI and available CT and/or MRI data between March 2008 and February 2018 at the Yamaguchi University Animal Medical Center. All general characteristics on the day of the first visit, including sex, age, breed, and weight, were required to be extractable from the medical record. All pet owners provided informed consent for the inclusion of their pets in this retrospective study. All dogs included in this study were diagnosed with AAI based on clinical symptoms, neurological examination, and subjective interpretation of the radiographic, MRI, and CT findings. Due to the nature of this study, which is particularly focused on the association with body weight, cases with only one breed of dog is excluded from the study.

## Analysis of the presence of concurrent diseases in dogs with AAI

Based on the findings of previous reports about veterinary imaging diagnostics, some abnormalities with clear diagnostic criteria were selected. Specifically, occipital dysplasia (OD), AOO, DD, CLM, SM, lateral ventricular enlargement (LVE), and intracranial arachnoid cyst (IAC) were examined. The diagnostic criteria for each disease have been explained below. Osirix Medical Imaging Software or Ziostation was used for image analysis.

## Occipital dysplasia (OD)

The N/h ratio was evaluated using CT imaging data according to a previous report by Baroni et al. [12]. The N/h ratio was calculated by assuming that the distance from the apex of the virtual line of the foramen magnum to the actual dorsal connection is N and the distance to the ventral side is h; OD was assumed to be 0.3 or more (Fig 1) [13].

## Atlanto-occipital overlapping (AOO)

AOO was evaluated using CT imaging data according to a previous report by Takahashi et al. [14]. In a 3-dimentional multi-planar reconstruction midsagittal section, the arcus dorsalis of

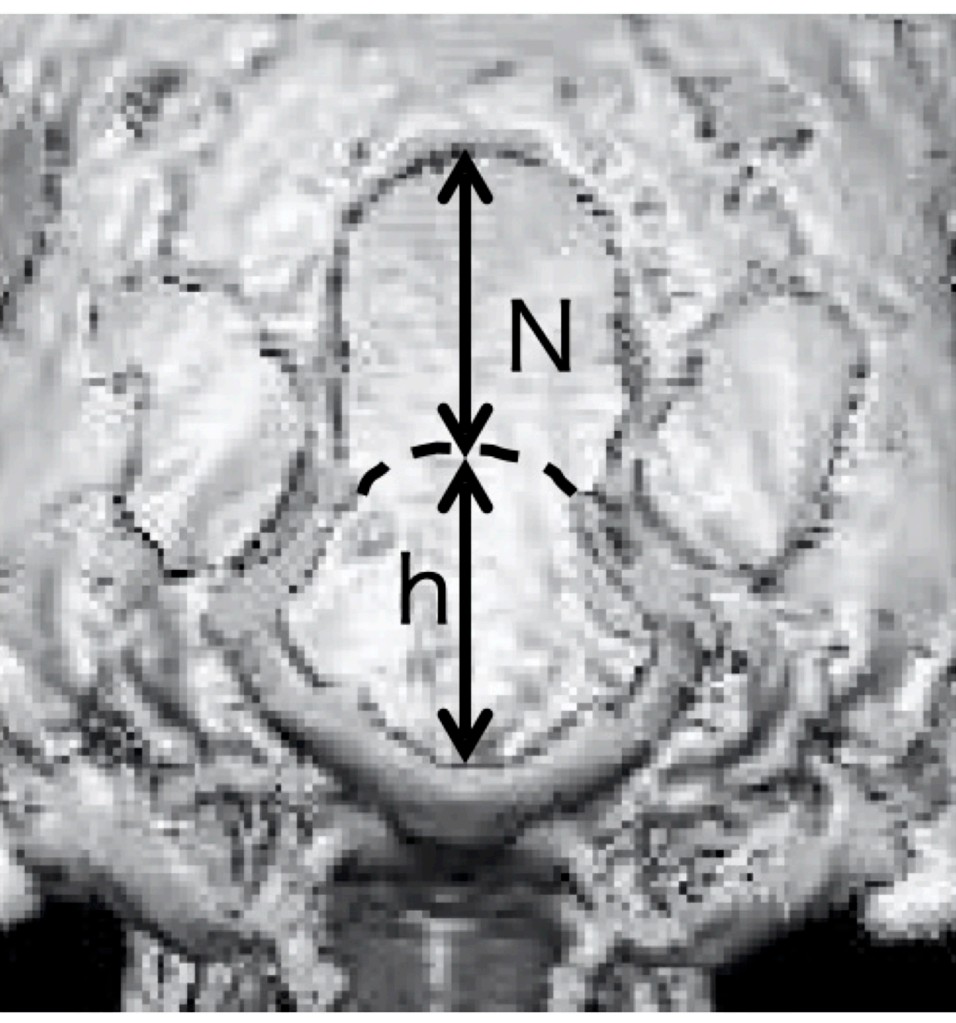

**Fig 1. Calculation of N/h ratio using CT image data.** The distance from the apex of the virtual line of the foramen magnum to the actual dorsal connection was calculated as N and the distance to the ventral side as h. The N/h ratio in this case was 0.96, and it was diagnosed as occipital dysplasia.

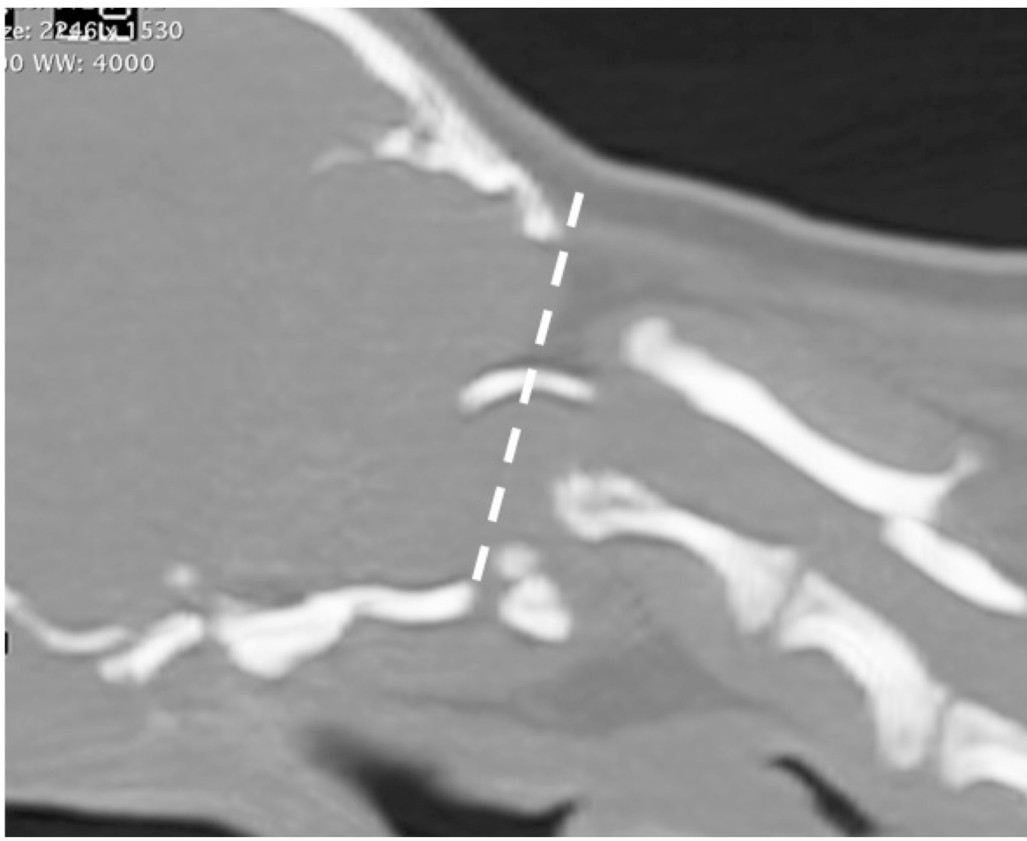

**Fig 2. Diagnosis of atlanto-occipital overlapping by CT imaging.** drawing the McRae line by connecting the opisthion and basion of the occipital bone in a three dimentional-multiplanar reconstruction mid-sagittal sectional view. atlanto-occipital overlapping was diagnosed when the arcus dorsalis of the atlas is located closer to the cranial aspect of the occipital bone than the McRae line.

the atlas is located closer to the cranial aspect of the occipital bone than the McRae line, which joins the opisthion and basion of the occipital bone (Fig 2).

### Dens dysplasia (DD)

A previous retrospective study reported that 76% (35/46) of dogs with AAI also had DD [15]. However, the diagnostic criteria for DD have not been defined clearly. Consequently, no clear diagnostic criteria for DD, except in cases with abnormal conformation/union or aplasia of dens, are available. Therefore, in this study, DD was diagnosed as abnormal conformation/union or aplasia of dens in the radiographs and CT images (Fig 3).

### Chiari-like malformation (CLM)

CLM was diagnosed according to the BVA guidelines for CM/SM. Cases with AOO were considered to not have CLM as the morphological evaluation was difficult due to the compression of the cerebellum by the dorsal side of the atlas.

### Syringomyelia (SM)

SM was diagnosed according to the BVA guidelines for CM/SM.

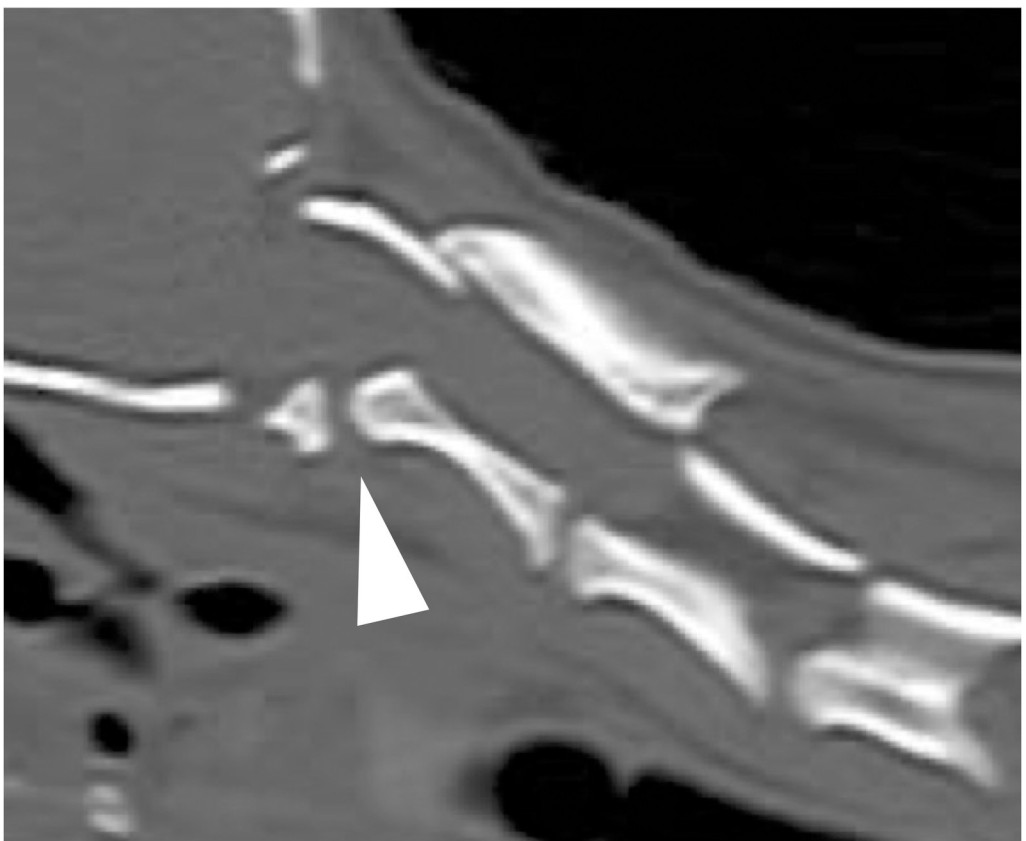

**Fig 3. A three dimentional-multiplanar reconstruction mid-sagittal sectional view of dens dysplasia.** There is no formation of the dens on the ventral side of the annulus.

### Lateral ventricular enlargement (LVE)

LVE was diagnosed based on a previous report by Wunschmann et al. [16]. The V/B ratio was calculated using the MRI data by considering the distance from the dorsal ventricular surface to the pituitary gland as B, and the height of the lateral ventricle as V. LVE was diagnosed when the V/B ratio exceeded 15% (Fig 4).

### Intracranial intra-arachnoid cyst (IAC)

IAC was diagnosed based on a previous report by Vernau et al. [17]. Using MRI data, IAC was diagnosed as an arachnoid cyst when excessive fluid retention was observed in the region of the corpora quadrigemina with sharply defined margins. The cyst contained fluid that was iso-dense to cere (Fig 5).

### Statistical analysis

Statistical analyses were performed using a commercially available software package (Graph-Pad Prism). A P-value of $< 0.05$ was considered statistically significant in all analyses. The two groups were compared using the *F*-test to determine homoscedasticity in each group. For equal variance, we performed an independent *t*-test, whereas for unequal variance, we performed Welch's test to compare the two groups. Pearson rank correlation coefficient was used to examine the relationship between body weight or age at onset and each continuous variable

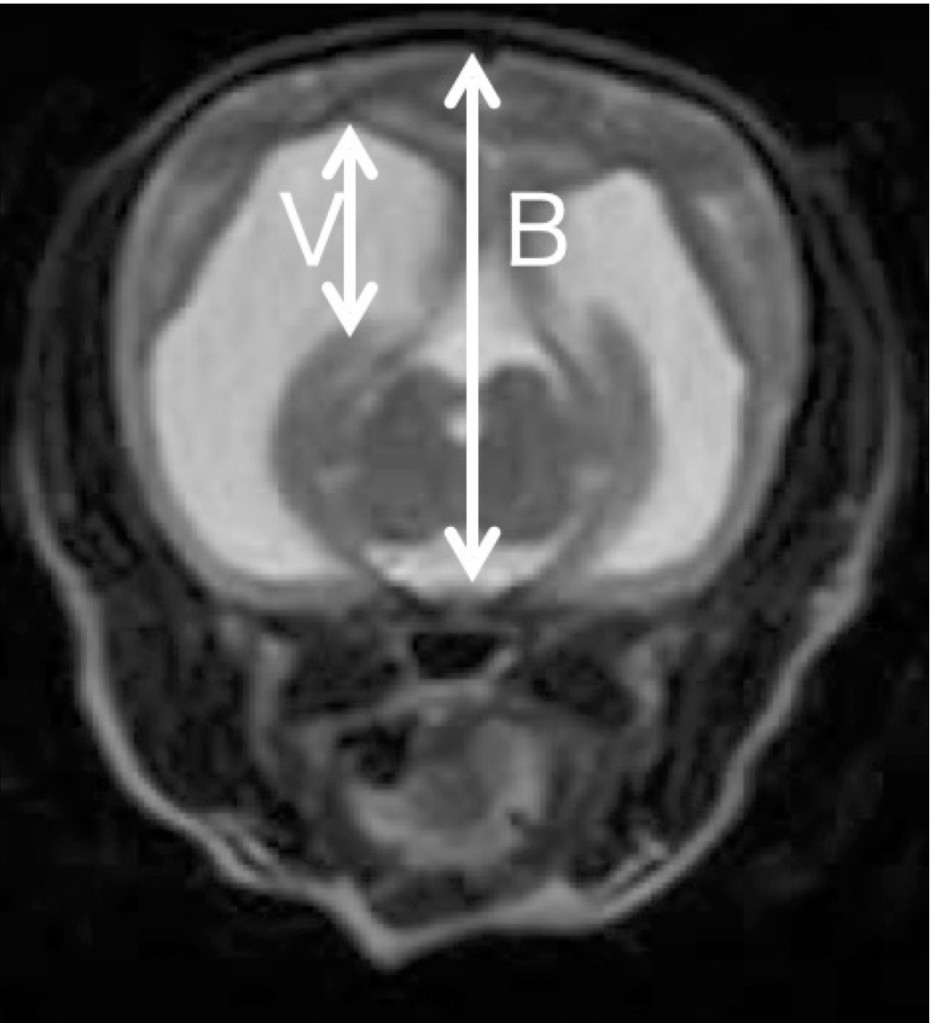

**Fig 4. A case diagnosed as lateral ventricular enlargement on magnetic resonance imaging.** The V/B ratio was calculated using B as the distance from the dorsal surface of the ventricle to the pituitary gland and V as the height of the lateral ventricle, and lateral ventricular enlargement was diagnosed when the V/B ratio was 15% or greater. In this case, the V/B ratio was 41%, and the patient was diagnosed as lateral ventricular enlargement.

measurement (the N/h and V/B ratios). Spearman rank correlation coefficient was used to examine the relationship between body weight or age at onset and the number of concurrent diseases.

## Results

### Animals

41 dogs, including 17 Chihuahuas, 10 Toy poodles, 4 Pomeranians, 3 Malteses, 3 Yorkshire terriers, 2 Papillons and 2 Miniature Dachshunds were participated in this study. One Italian Greyhound, Pug and Cavalier King Charles Spaniel were each selected, but they were excluded from this study in accordance with the inclusion criteria. 23 dogs were males (including 3 castrated males), and 18 dogs were females (including 3 spayed females). The median age at onset was 290.00 (30–3886) days [700.07 ± 938.27 (mean±standard deviation)], and the median body weight was 1.89 (0.92–4.95) kg [2.15±1.01] (Table 1).

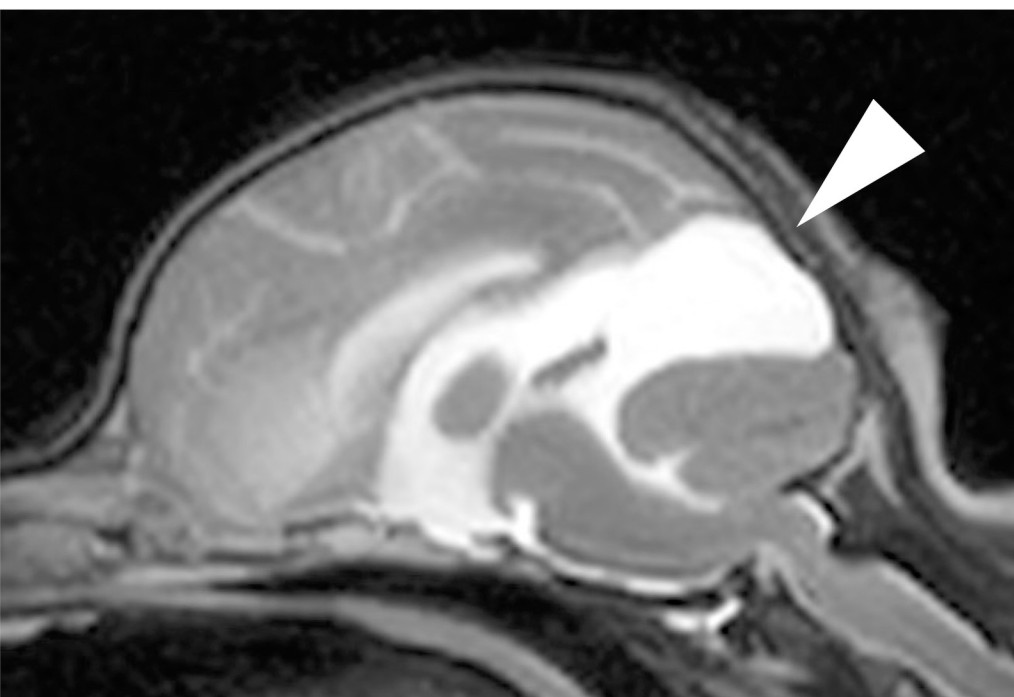

**Fig 5. The patient was diagnosed as intracranial arachnoid cyst by magnetic resonance imaging.** Specifically, excessive fluid retention in the corpora quadrigem region with sharply defined margins and containing cyst fluid isodense to cerebrospinal fluid were observed.

## Occipital dysplasia (OD)

The medical records confirmed that no surgical procedures, such as foramen magnum decompression, had been performed in the occipital bone region. In addition, it was also assumed that this study could be performed regardless of the timing of the imaging. 27 dogs with 3D-MPR images of the occipital bone were included based on their previous data. 17 dogs were diagnosed with OD via reconstruction of the 3D-MPR data. Comparison of body weight

**Table 1. Characteristics of study population (n = 41).**

| Breed (n) | Chihauha | 17 |
|---|---|---|
| | Toy poodle | 10 |
| | Pomeranian | 4 |
| | Yorkshire terrier | 3 |
| | Maltese | 3 |
| | Papillon | 2 |
| | Minichua dachshound | 2 |
| Sex (n) | Intact Male | 23 |
| | Castrated male | 3 |
| | Intact female | 18 |
| | Spayed female | 3 |
| Age (days) | Mean ± SD | 700.07 ± 938.27 |
| | Median (range) | 290 (30–3886) |
| Weight (kg) | Mean ± SD | 2.15 ± 1.01 |
| | Median (range) | 1.89 (0.92–4.95) |

**Table 2. Comparison about age at onset, and body weight of each disease presence and absence.**

| Diagnosis | n | positive | negative | Age at onset (days) (Average ± SD) | | | | | | P value | Weight (kg) (Average ± SD) | | | | | | P value |
|---|---|---|---|---|---|---|---|---|---|---|---|---|---|---|---|---|---|
| | | | | positive | | | negative | | | | positive | | | negative | | | |
| Occipital dysplasia | 27 | 17 | 10 | 726.65 | ± | 1030.64 | 276.10 | ± | 170.19 | 0.1054 | 1.89 | ± | 0.90 | 2.79 | ± | 1.10 | 0.0547 |
| AOO | 34 | 22 | 12 | 816.45 | ± | 950.86 | 228.75 | ± | 170.53 | 0.0113* | 1.78 | ± | 0.71 | 2.71 | ± | 1.15 | 0.0269* |
| Dens dysplasia | 41 | 27 | 14 | 435.48 | ± | 549.13 | 1210.36 | ± | 1240.29 | 0.0476* | 2.29 | ± | 1.14 | 1.86 | ± | 0.56 | 0.1226 |
| Chiari-like malfomation | 34 | 8 | 26 | 714.25 | ± | 1213.11 | 726.58 | ± | 900.67 | 0.9806 | 2.08 | ± | 1.05 | 2.03 | ± | 0.80 | 0.9093 |
| Syringomyaria | 34 | 5 | 29 | 1749.00 | ± | 1245.46 | 542.36 | ± | 822.24 | 0.1252 | 1.86 | ± | 0.59 | 1.99 | ± | 0.81 | 0.7049 |
| Lateral ventricles enlargement | 36 | 20 | 16 | 888.25 | ± | 1191.63 | 443.25 | ± | 459.64 | 0.1476 | 1.60 | ± | 0.40 | 2.75 | ± | 1.08 | 0.001* |
| Intracranial arachnoid cyst | 35 | 5 | 30 | 2198.80 | ± | 928.83 | 457.50 | ± | 728.42 | 0.0173* | 2.01 | ± | 0.38 | 2.03 | ± | 0.90 | 0.9358 |

and age at onset between the groups with and without OD showed no significant difference (726.65 ± 1030.64 vs 276.1 ± 170.19 days, P = 0.1054) (1.89 ± 0.9 vs 2.79 ± 1.1 kg, P = 0.0547) (Table 2). The correlation between the N/h ratio, age at onset, and body weight was examined using Peason's method; however, no significant differences were observed (R = -0.1825, P = 0.4833) (R = -0.2664, P = 0.3013) (Fig 6).

## Atlanto-occipital overlapping (AOO)

34 dogs whose 3D-MPR or mid-sagittal cross-sectional CT images of the occipital bone region acquired at the time of initial examination were available were included in this study.22 dogs were diagnosed with AOO. Comparison of body weight and age at onset of disease between the groups with and without AOO revealed that the age at onset was significantly higher in the group with AOO (816.45 ± 950.86 vs 228.75 ± 170.53 days, P = 0.0133) and body weight was significantly lower in the group with AOO (1.78 ± 0.71 vs 2.71 ± 1.15 kg, P = 0.0269) (Table 2).

## Dens dysplasia (DD)

41 dogs whose 3D-MPR images or mid-sagittal cross-sectional images of cervical CT or cervical radiography acquired at the time of initial examination were available were included in this study. 27 dogs were diagnosed with DD. Comparison of body weight and age at onset between the groups with and without DD revealed that the age at onset was significantly lower in the group with DD (435.48 ± 549.13 vs 1210.36 ± 1240.29 days, P = 0.0476). No significant differences were observed in body weight (2.29 ± 1.14 vs 1.86 ± 0.56 kg, P = 0.1226) (Table 2).

## Chiari-like malformation (CLM)

MRI scanning of the brain was performed at the time of initial diagnosis in 34 dogs such that the morphology of the cerebellum could be evaluated in the sagittal section. There were 8 dogs diagnosed as CLM. No significant differences were observed in the age at onset or body weight (714.25 ± 1213.11 vs 726.58 ± 900.67 days, P = 0.9806) (2.08 ± 1.05 vs 2.03 ± 0.80 kg, P = 0.9093) (Table 2).

## Syringomyelia (SM)

MRI scanning of the cervical region was performed at the time of initial diagnosis in 34 dogs such that the morphology of the cervical spinal cord could be evaluated in the sagittal section. There were 5 dogs diagnosed SM. No significant differences were observed in the age at onset or body weight (1749.00 ± 1245.46 vs 542.36 ± 822.24 days, P = 0.1252) (1.86 ± 0.59 vs 1.99 ± 0.81 kg, P = 0.7049) (Table 2).

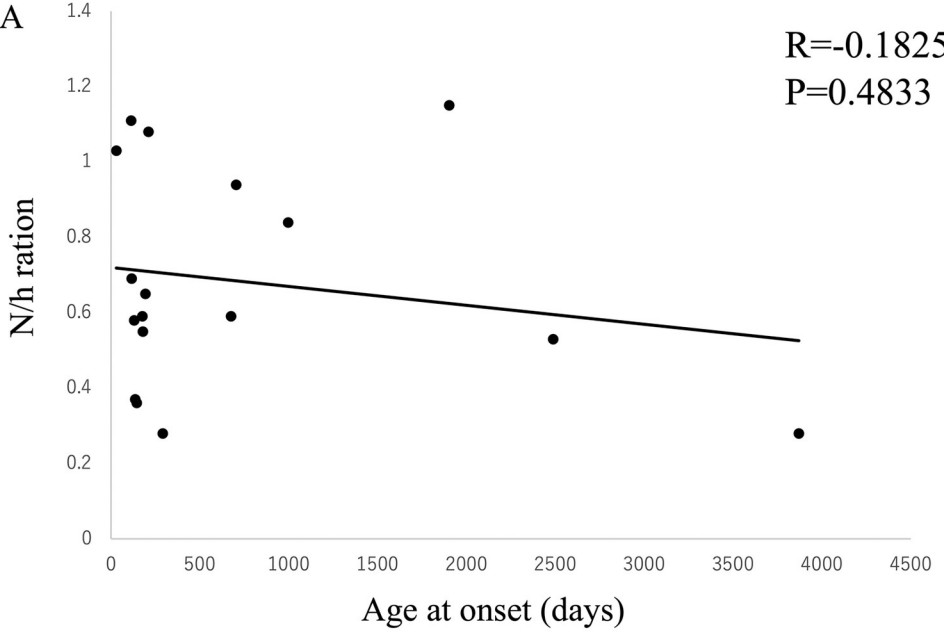

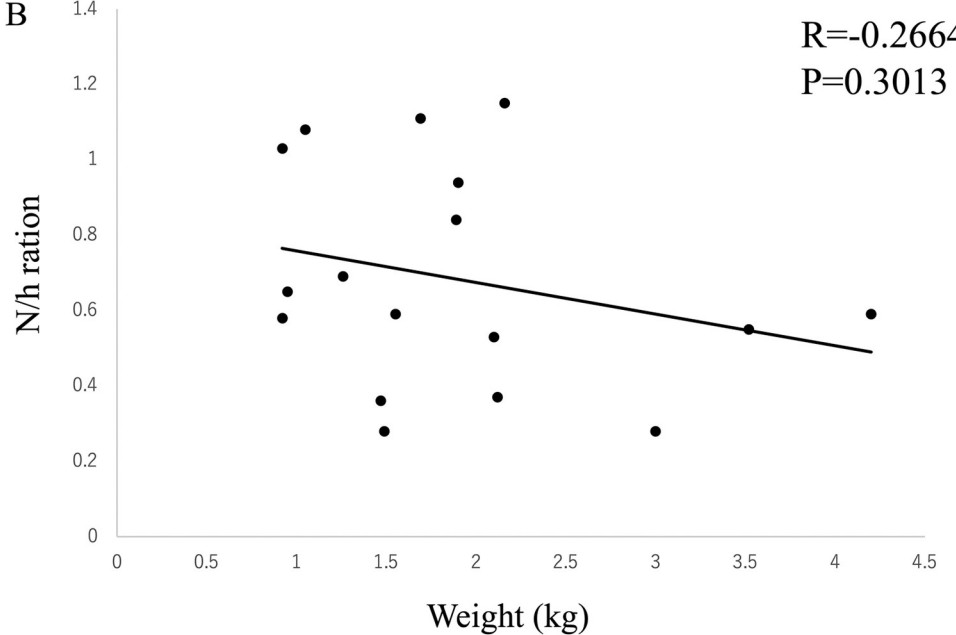

**Fig 6.** Correlations between N/h ratio, (A) age at onset, and (B) body weight were examined, and no significant differences were observed.

### Lateral ventricle enlargement (LVE)

36 dogs whose lateral ventricles could be evaluated in the brain MRI or CT images were included in this study. Among these 36 dogs, 20 were diagnosed with LVE based on the criteria used in previous reports. Comparison of body weight and the age at onset between the groups with and without LVE revealed that the group with LVE had significantly lower body weight (888.25 ± 1191.63 vs 443.25 ± 459.64 days, P = 0.1476) (1.60 ± 0.40 vs 2.75 ± 1.08 kg,

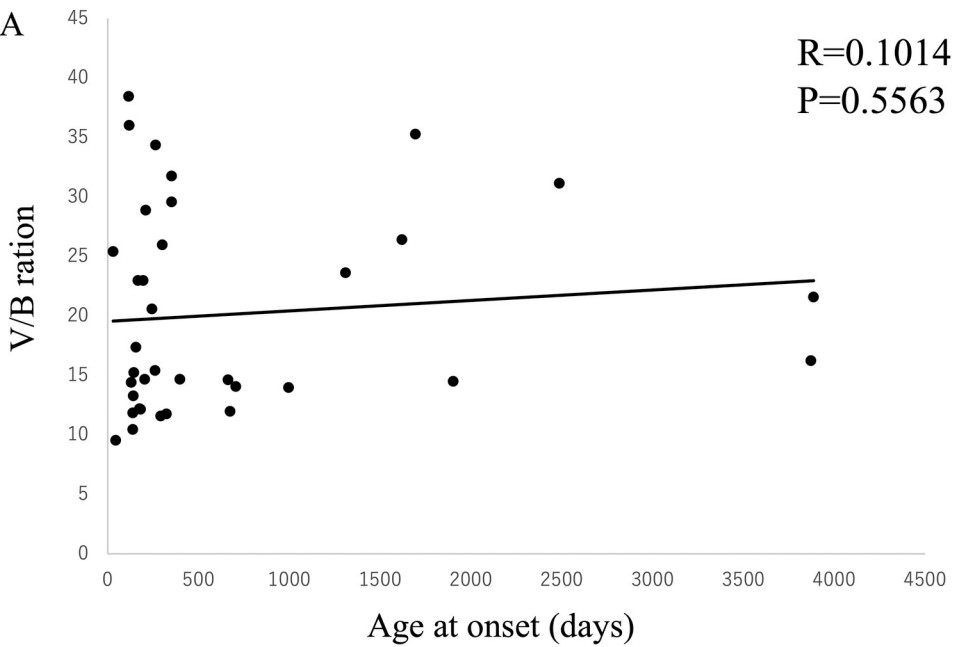

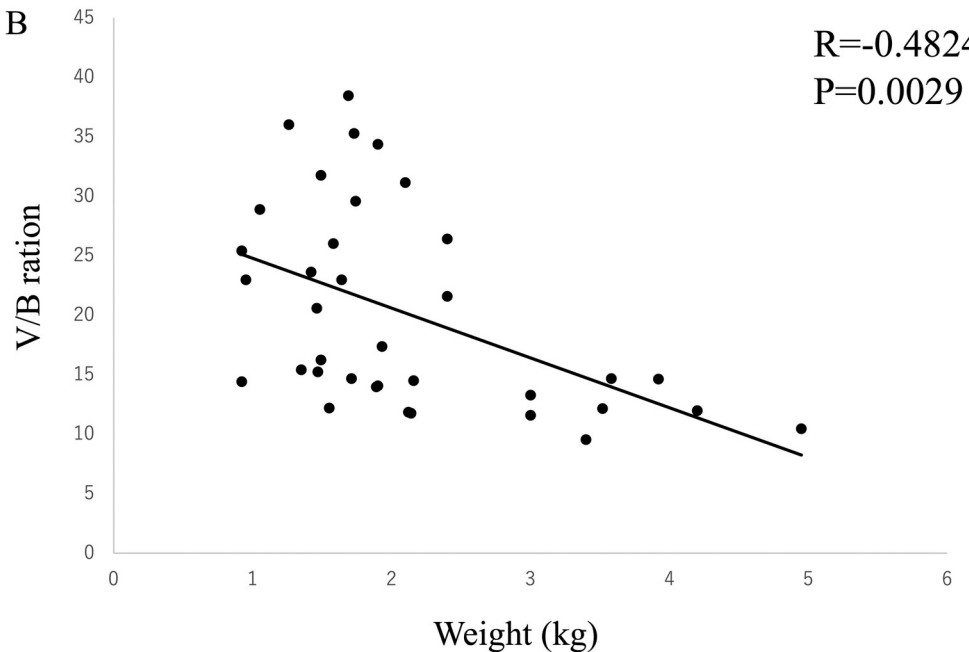

**Fig 7.** Correlations between the V/B ratio, (A) age at onset and, and (B) body weight were examined. No correlation was found between the age at onset and the V/B ratio, but a significant correlation was found between the body weight and the V/B ratio.

P = 0.001) (Table 2). Correlation analysis of the V/B ratio, age at onset, and weight using Pearson's method revealed that there was no correlation between the age at onset and V/B ratio (R = 0.1014, P = 0.5563); however, a significant correlation was observed between body weight and the V/B ratio (R = -0.4824, P = 0.0029) (Fig 7).

### Intracranial intra-arachnoid cyst (IAC)

MRI scanning of the head was performed at the time of initial examination in 35 dogs such that the occipital morphology could be evaluated in the sagittal section. These dogs were included in this study. IAC was observed in 5 dogs. Comparison of body weight and the age at onset between the groups with and without IAC revealed that the age at onset was significantly higher in the group with IAC (2198.80 ± 928.83 vs 457.5 ± 728.42 days, P = 0.0173). No significant differences were observed in body weight (2.01 ± 0.38 vs 2.03 ± 0.90 kg, P = 0.9358) (Table 2).

### Number of concurrent diseases

Spearman's method was used to evaluate the correlation between the total number of concurrent diseases and the age at onset and body weight in all patients (n = 25) with all imaging findings for all the concurrent diseases studied in this study (OD, AOO, DD, CLM, SM, LVE, and IAC). There was no correlation with age at onset (R = 0.0758, P = 0.7188); however, it was observed that the lower the body weight, the higher the number of concurrent diseases (R = -0.5037, P = 0.0103) (Fig 8).

## Discussion

This study analyzed the relationship between the presence of concurrent diseases and clinical parameters in patients with AAI. The results showed that most patients with AAI had concurrent diseases, and the body weight of those with AOO and LVE tended to be lower than that of those without these conditions. Furthermore, when the number of concurrent diseases and clinical parameters were compared between patients with OD, AOO, DD, CLM, SM, LVE, and IAC, a correlation was observed between body weight and the number of concurrent diseases.

 CJA is a syndrome caused by congenital dysplasia of the region extending from the occipital bone to the upper cervical vertebrae. When first described, CJA included CLM, atlanto-occipital instability, AAI, occipital-atlanto-axial dysplasia, AOO, and dorsal spinal cord compression. However, the pathogenesis and primary causes of CJA have not been clarified [10]. This study examined toy-breed dogs (Chihuahuas, Toy Poodles, Pomeranians, Yorkshire Terriers, Papillons, Maltese, and Miniature Dachshunds) diagnosed with AAI at the Yamaguchi university and analyzed the following abnormal findings in the head and spinal cord region that were clearly recognizable on images and had been reported as diagnostic imaging findings in the past: OD, AOO DD, CLM, SM, LVE, and IAC. Among the items under consideration, those recognized in previous reports as categories of CJA include CLM, AAI, AOO, DD, and SM. LVE is reportedly associated with CLM and other cervical CSF circulatory disturbances in humans and may also be present in dogs; however, its detailed pathogenesis has not been elucidated [18, 19]. In addition, the pathogenesis of IAC and its relationship with CJA have not been clarified. This study examined these two diseases and the other diseases included in CJA that were clearly recognizable via imaging. Multiple imaging abnormalities were found as a result, indicating that AAI is associated with multiple concurrent diseases.

 In this study, AOO was observed in 22/34 dogs, and the body weight of the dogs with AOO tended to be lower than that of the dogs without AOO. AOO is thought to cause clinical symptoms by penetrating the dorsal side of the atlas through the greater occipital foramen and compressing the cerebellum. Moreover, it reportedly affects the clinical signs and prognosis of dogs with AAI [20]. Previous reports have shown no significant differences in the age at onset in the group with AOO compared with that of the group without AOO; however, a significant difference was observed in body weight, and no breed restriction was established in this study [11]. Since this study examined toy-breed dogs, the results showing that the body weight of

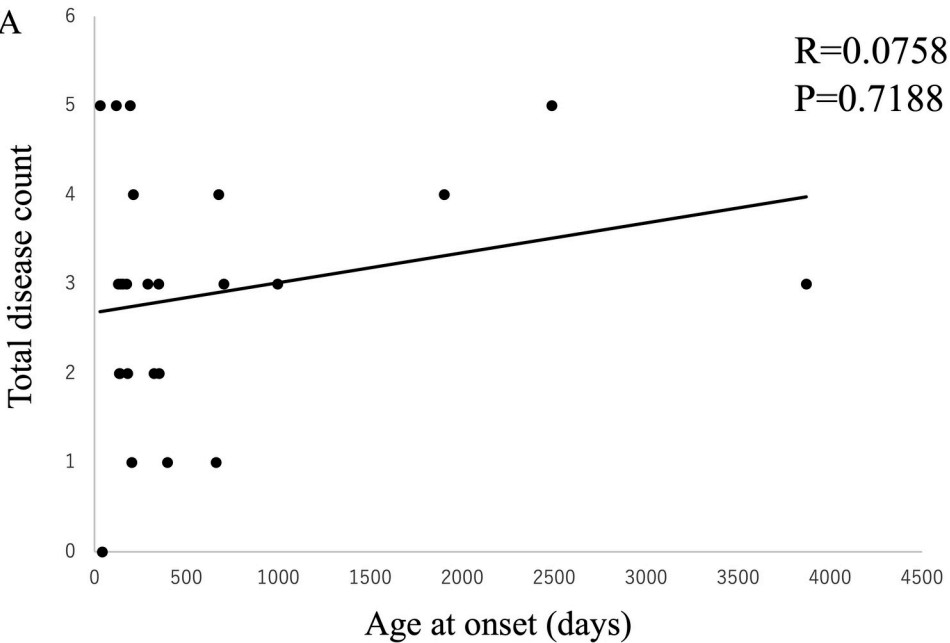

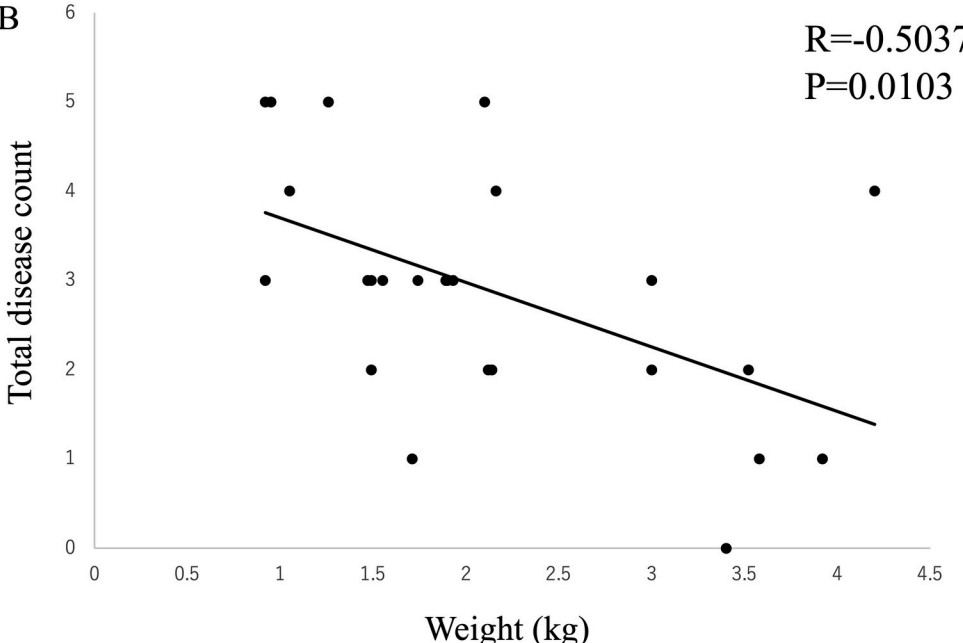

**Fig 8.** Correlations between the total number of concurrent diseases, (A) age at onset, and (B) body weight were examined. No correlation was found between the age at onset and the number of concurrent diseases; however, body weight tended to be lower in patients with higher number of concurrent diseases.

patients with AOO tended to be lower than that of those without AOO may be more accurate regarding weight and AOO prevalence.

In this study, DD was observed in 27/41 (about 66%) dogs. A previous report showed that 35/46 (76%) dogs with AAI had DD [15]. The low prevalence of DD in this study may be due

to only cases with radiographically evident aplasia or ossification failure being considered DD. No veterinary criteria are available for the diagnosis of DD at present. Takahashi et al. measured the dens-to-axis length ratio (DALR) and dens angle (DA) and compared them between dogs with and without AAI [21]. In addition to the large differences between breeds, these values do not provide an objective index that defines the level of DD [14]. Therefore, we did not measure these values in this study. However, based on these results and the results of the present study, it is possible that an association may be present between body weight and DALR and DA, and this association should be explored in future studies along with the diagnostic method for DD.

Dens ossification of the axial vertebral body occurs at around 7–9 months of age; therefore, it is possible that images acquired before ossification in young patients are incorporated into DD. In this study, the age at onset of the dogs with DD was significantly lower than that of dogs without DD, suggesting that ossification may progress during development.

CLM was observed in 8/34 dogs in this study, and no significant difference was observed in the age at onset and weight. In previous reports, no significant difference was observed in the age at Cavalier King Charles Spaniels with/without CLM and SM [22]. In this study, the cases with AOO were included in the group without CLM due to difficulty in evaluating the morphology of the cerebellum in such cases. Moreover, it has been reported in the past that a certain percentage of patients diagnosed with CLM using MRI have AOO; however, it is difficult to determine the actual prevalence of AOO as its pathogenesis is unknown. SM is associated with CLM and other CSF circulatory disorders, and the incidence of SM is reported to be higher in those with AOO (28.2% vs. 11%) [11]. In our study, all cases with SM also had AOO or CLM.

In this study, the body weight of the dogs in the group with LVE tended to be lower than that of those in the group without LVE, and the V/B ratio and body weight showed a negative correlation. Disturbances in the circulation of CSF at the medulla oblongata-cervical junction, as observed in AAI, causes enlargement of the central canal of the spinal cord and lateral ventricles [20]. Moreover, the lateral ventricles are less compliant than the third and fourth ventricles and have a higher rate of volume change in response to circulatory disturbances in humans [23]. It has been suggested that impaired CSF circulation at the medulla oblongata-cervical junction is associated with the development of secondary hydrocephalus in dogs [24, 25]. Based on the results of previous studies and those of the present study, enlargement of the lateral ventricles may be a result of impaired CSF circulation associated with AAI, and this effect may be more pronounced in dogs with lower body weight.

The number of concurrent diseases and body weight showed a negative correlation in this study; however, no significant difference was observed in the age at onset. Previous reports have shown that the body weight of patients with AOO tend to be lower than that of those without AOO [11]. However, no reports have examined other diseases, and this is the first report of its kind. Although CJA occurs in toy-breed dogs, our results suggest that smaller breeds are more likely to develop complex malformations caused by CJA.

Ventral fixation of the atlanto-axial joint shows a relatively good prognosis in patients with AAI [26, 27]. However, some cases do not show any postoperative neurological improvement, and in some cases, worsening of the symptoms is observed. This is an observational study, and it is not revealed whether lighter body weight affects prognosis by our study. However, Some reports suggest that dorsal decompression should be considered based on preoperative CT or MRI imaging [20, 24]. In contrast, other reports suggest that impaired cerebrospinal fluid (CSF) circulation in the cervical region is involved in the development of syringomyelia (SM) and hydrocephalus in dogs with CJA [25]. Based on these previous reports and our study, a detailed imaging diagnosis of the concurrent diseases and analysis of clinical symptoms and

neurological prognosis may provide useful information for treatment of AAI, selection of surgical procedures, and identification of poor prognosis patients.

This study had some limitations. First, the patients in this study were treated by several veterinarians, and the imaging diagnosis of AAI was based on the subjective diagnosis of each veterinarian. Second, it was a retrospective study conducted on the basis of information extracted from the medical records of the participants; therefore, it was difficult to evaluate the prognosis of cases with many imaging complex malformations. Third, since metal implants were used in many cases, MRI evaluation of the cerebrospinal cord was not possible in many cases, and it was unclear whether the CSF circulatory disturbance was improved by the surgical procedure. Lastly, since the definition of CJA remains unclear, it is possible that other conditions were incorporated into the criteria for this disease.

In conclusion, the present study demonstrated that among the cases with AAI, those with AOO and LVE tended to have lower body weight. Moreover, lower body weight was associated with a higher number of concurrent diseases. These results suggest that AAI is one of the complex malformations observed in toy-breed dogs that need to be recognized. Thus, additional examination and treatment methods must be considered carefully, especially in cases with low body weight.

## Supporting information

**S1 Data.**
(XLSX)

## Author Contributions

**Conceptualization:** Harumichi Itoh, Kazuhito Itamoto.

**Investigation:** Harumichi Itoh.

**Methodology:** Harumichi Itoh.

**Project administration:** Takuya Itamoto, Kazuhito Itamoto.

**Supervision:** Kenji Tani, Hiroshi Sunahara, Yuki Nemoto, Munekazu Nakaichi, Toshie Iseri, Hiro Horikirizono, Kazuhito Itamoto.

**Writing – original draft:** Harumichi Itoh.

**Writing – review & editing:** Harumichi Itoh.

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
