## [Decision Letter · Decision Letter 0]

21 Jun 2023

PONE-D-23-12731Relationship between clinical parameters and malformations in patients diagnosed with atlanto-axial instabilityPLOS ONE

Dear Dr. Itoh,

Thank you for submitting your manuscript to PLOS ONE. After careful consideration, we feel that it has merit but does not fully meet PLOS ONE’s publication criteria as it currently stands. Therefore, we invite you to submit a revised version of the manuscript that addresses the points raised during the review process.

Please see the reviewer comments below. Please pay particular attention to the title and ensure that "patient" is removed and that it is clear that the study is performed in dogs - it appears that this already caused confusion in the review process.  Please also update your data availability statement to include information on where others can access the data, as data must be available to other researchers as per PLOS ONE publication criteria.

We look forward to receiving your revised manuscript.

Kind regards,

Hanna Landenmark

Staff Editor

PLOS ONE

3. Please include your tables as part of your main manuscript and remove the individual files. Please note that supplementary tables (should remain/ be uploaded) as separate "supporting information" files

Reviewers' comments:

Reviewer's Responses to Questions

**Comments to the Author**

1. Is the manuscript technically sound, and do the data support the conclusions?

Reviewer #1: Partly

Reviewer #2: Partly

2. Has the statistical analysis been performed appropriately and rigorously? 

Reviewer #1: No

Reviewer #2: Yes

3. Have the authors made all data underlying the findings in their manuscript fully available?

Reviewer #1: Yes

Reviewer #2: Yes

4. Is the manuscript presented in an intelligible fashion and written in standard English?

Reviewer #1: Yes

Reviewer #2: Yes

5. Review Comments to the Author

Reviewer #1: Thank you for allowing me to review this manuscript. The authors present an interesting retrospective study that identifies the co-morbid frequency of what can loosely be called craniovertebral junction anomalies in the presence of atlanto-axial instability in a subset of toy breeds. The methodology as written has some weaknesses, but this may be due to the description rather than structural.

Specific comments:

• Abstract Line 29-30: The second sentence in the Abstract is incomplete and does not make sense as written.

• Abstract Line 38-39: Consider including % difference in body weight between groups

• Abstract Line 43: ‘…especially in the case of small dogs.’ is not precise given the study is on toy breeds.

• Introduction Line 71-80: The point of this paragraph is not clear.

• Case Selection Line 94-98: The inclusion/exclusion criteria are not clear. Were the listed breeds the only ones found in the records review or were they selected before the review for some other reasons? Was it a sample of convenience? King Charles Cavalier Spaniels are mentioned in the discussion but none are included in the analysis.

• Case Selection: Please expand on the diagnostic criteria (or findings) that were used for AAI in the records review.

• General Findings Line 100: Using the header ‘General Findings’ here makes it sound like you are reporting results.

• Lines 105-112: This background does not belong in the Methods section.

• Lines 121-122: This Fig. 1 reference does not make sense as it refers to correlation with age at onset and weight, and then reports results. Results should not be reported in the Methods section.

• Lines 150-152: This Fig. 2 reference does not make sense for the same reasons as cited above.

• Statistics

o What is the rationale for using Mann-Whitney instead of t-tests? Were the ages/weights tested for normality? One would expect weights to be normally distributed.

o The significance level should be adjusted for multiple observations/tests, for example with a Bonferroni correction.

• Results:

o Throughout the entire results section p values should be included when relationships are noted.

o It would also be helpful to provide some means and/or % differences in the weight and age so the reader has a sense of the effect size (or lack thereof)

o For the ‘Number of concurrent diseases’ provide the actual r and p values in the Results section.

• Discussion: It is not clear why the CKCS is discussed.

Reviewer #2: 1、 The topic of this paper should mention that this study is made in dogs

2、The paper has investigated several diseases related with AAI. And found the relationship of body weights and age with the occureences of different diseases of AAI. This is only the results of observation and lack of analysis of mechanisms. And in addition, by only observations of age and bodyweights, the results can not give readers of this paper any help for the prognosis of AAI.

6. PLOS authors have the option to publish the peer review history of their article (what does this mean?). If published, this will include your full peer review and any attached files.

Reviewer #1: No

Reviewer #2: No

---

## [Author Response · Author response to Decision Letter 0]

6 Aug 2023

Reviewer #1: Thank you for allowing me to review this manuscript. The authors present an interesting retrospective study that identifies the co-morbid frequency of what can loosely be called craniovertebral junction anomalies in the presence of atlanto-axial instability in a subset of toy breeds. The methodology as written has some weaknesses, but this may be due to the description rather than structural. Specific comments:

 • Abstract Line 29-30: The second sentence in the Abstract is incomplete and does not make sense as written.

We modified as below.

Furthermore, patients with atlanto-axial instability are predisposed to other concurrent diseases. 

 • Abstract Line 38-39: Consider including % difference in body weight between groups

We modified as below.

The body weight of the patients in the groups with atlanto-occipital overlapping and lateral ventricular enlargement was lower than that of those in the groups without these diseases (1.78 ± 0.71 vs 2.71 ± 1.15 kg, P=0.0269, 1.60 ± 0.40 vs 2.75 ± 1.08 kg, P=0.001, respectively). 

 • Abstract Line 43: ‘…especially in the case of small dogs.’ is not precise given the study is on toy breeds.

We removed this sentence.

 • Introduction Line 71-80: The point of this paragraph is not clear.

This paragraph is a description of the treatment and prognosis of AAI. As pointed, we have determined that it is not appropriate for introduction of this paper. However, since the results of this study suggest that body weight may play a role in the choice of treatment and prognosis, some of the content has been moved to the discussion section.

 • Case Selection Line 94-98: The inclusion/exclusion criteria are not clear. Were the listed breeds the only ones found in the records review or were they selected before the review for some other reasons? Was it a sample of convenience? King Charles Cavalier Spaniels are mentioned in the discussion but none are included in the analysis.

We modified case selection as below

We retrospectively analyzed dogs diagnosed with AAI and available CT and/or MRI data between March 2008 and February 2018 at the Yamaguchi University Animal Medical Center. All pet owners provided informed consent for the inclusion of their pets in this retrospective study. All dogs included in this study were diagnosed with AAI based on clinical symptoms, neurological examination, and subjective interpretation of the radiographic, MRI, and CT findings. Due to the nature of this study, which is particularly focused on the association with body weight, cases with only one breed of dog is excluded from the study.

In addition, a previous report about relationship between AOO, one of concurrent diseases of AAI and body weight is described in the introduction. We have also included more detailed information on the results.

 • Case Selection: Please expand on the diagnostic criteria (or findings) that were used for AAI in the records review.

We modified as below

All dogs included in this study were diagnosed with AAI based on clinical symptoms, neurological examination, and subjective interpretation of the radiographic, MRI, and CT findings. 

Also, we added a new sentence in discussion. 

This study had some limitations. First, the patients in this study were treated by several veterinarians, and the imaging diagnosis of AAI was based on the subjective diagnosis of each veterinarian. 

 • General Findings Line 100: Using the header ‘General Findings’ here makes it sound like you are reporting results.

We removed this paragraph and added a sentence in case selection as below.

All general characteristics on the day of the first visit, including sex, age, breed, and weight, were required to be extractable from the medical record. 

 • Lines 105-112: This background does not belong in the Methods section.

We modified this sentences as below

Based on the findings of previous reports about veterinary imaging diagnostics, some abnormalities with clear diagnostic criteria were selected. Specifically, occipital dysplasia (OD), AOO, DD, CLM, SM, lateral ventricular enlargement (LVE), and intracranial arachnoid cyst (IAC) were examined. The diagnostic criteria for each disease have been explained below. Osirix Medical Imaging Software or Ziostation was used for image analysis.

 • Lines 121-122: This Fig. 1 reference does not make sense as it refers to correlation with age at onset and weight, and then reports results. Results should not be reported in the Methods section.

• Lines 150-152: This Fig. 2 reference does not make sense for the same reasons as cited above.

We removed these sentences

 • Statistics o What is the rationale for using Mann-Whitney instead of t-tests? Were the ages/weights tested for normality? One would expect weights to be normally distributed.

Statistical analysis was performed again according to be pointed out. In addition, the comparative analysis between the two groups was modified as follows.

The two groups were compared using the F-test to determine homoscedasticity in each group. For equal variance, we performed an independent t-test, whereas for unequal variance, we performed Welch's test to compare the two groups. 

 o The significance level should be adjusted for multiple observations/tests, for example with a Bonferroni correction.

This study did not perform multiple observations.

 • Results: o Throughout the entire results section p values should be included when relationships are noted.

We added p values for all data with significance. 

 o It would also be helpful to provide some means and/or % differences in the weight and age so the reader has a sense of the effect size (or lack thereof)

We added mean values for all data.

 o For the ‘Number of concurrent diseases’ provide the actual r and p values in the Results section.

We added R and P value of each results.

 • Discussion: It is not clear why the CKCS is discussed.

We removed these sentences.

Reviewer #2: 1、 

The topic of this paper should mention that this study is made in dogs

We modified our title as “Relationship between clinical parameters and malformations in dogs diagnosed with atlanto-axial instability”

 2、The paper has investigated several diseases related with AAI. And found the relationship of body weights and age with the occureences of different diseases of AAI. This is only the results of observation and lack of analysis of mechanisms. And in addition, by only observations of age and bodyweights, the results can not give readers of this paper any help for the prognosis of AAI.

We modified discussion as below

Ventral fixation of the atlanto-axial joint shows a relatively good prognosis in patients with AAI [25, 26]. However, some cases do not show any postoperative neurological improvement, and in some cases, worsening of the symptoms is observed. This is an observational study, and it is not revealed whether lighter body weight affects prognosis by our study. However, Some reports suggest that dorsal decompression should be considered based on preoperative CT or MRI imaging [19, 23]. In contrast, other reports suggest that impaired cerebrospinal fluid (CSF) circulation in the cervical region is involved in the development of syringomyelia (SM) and hydrocephalus in dogs with CJA [24]. Based on these previous reports and our study, a detailed imaging diagnosis of the concurrent diseases and analysis of clinical symptoms and neurological prognosis may provide useful information for treatment of AAI, selection of surgical procedures, and identification of poor prognosis patients.

This study had some limitations. First, the patients in this study were treated by several veterinarians, and the imaging diagnosis of AAI was based on the subjective diagnosis of each veterinarian. Second, it was a retrospective study conducted on the basis of information extracted from the medical records of the participants; therefore, it was difficult to evaluate the prognosis of cases with many imaging complex malformations. Third, since metal implants were used in many cases, MRI evaluation of the cerebrospinal cord was not possible in many cases, and it was unclear whether the CSF circulatory disturbance was improved by the surgical procedure. Lastly, since the definition of CJA remains unclear, it is possible that other conditions were incorporated into the criteria for this disease.

---

## [Decision Letter · Decision Letter 1]

2 Oct 2023

PONE-D-23-12731R1Relationship between clinical parameters and malformations in dogs diagnosed with atlanto-axial instabilityPLOS ONE

Dear Dr. Itoh,

Thank you for submitting your manuscript to PLOS ONE. After careful consideration, we feel that it has merit but does not fully meet PLOS ONE’s publication criteria as it currently stands. Therefore, we invite you to submit a revised version of the manuscript that addresses the points raised during the review process.

We look forward to receiving your revised manuscript.

Kind regards,

Miquel Vall-llosera Camps

Staff Editor

PLOS ONE

Journal Requirements:

**Additional Editor Comments:**

Please address Reviewer#2's remaining concerns in the attached document.

Reviewers' comments:

Reviewer's Responses to Questions

**Comments to the Author**

1. If the authors have adequately addressed your comments raised in a previous round of review and you feel that this manuscript is now acceptable for publication, you may indicate that here to bypass the “Comments to the Author” section, enter your conflict of interest statement in the “Confidential to Editor” section, and submit your "Accept" recommendation.

Reviewer #1: All comments have been addressed

Reviewer #2: All comments have been addressed

2. Is the manuscript technically sound, and do the data support the conclusions?

Reviewer #1: (No Response)

Reviewer #2: Yes

3. Has the statistical analysis been performed appropriately and rigorously? 

Reviewer #1: (No Response)

Reviewer #2: Yes

4. Have the authors made all data underlying the findings in their manuscript fully available?

Reviewer #1: (No Response)

Reviewer #2: Yes

5. Is the manuscript presented in an intelligible fashion and written in standard English?

Reviewer #1: (No Response)

Reviewer #2: Yes

6. Review Comments to the Author

Reviewer #1: (No Response)

Reviewer #2: The authors made all data underlying the findings in their manuscript fully available. The manuscript describe a technically sound piece of scientific research with data that supports the conclusions.

7. PLOS authors have the option to publish the peer review history of their article (what does this mean?). If published, this will include your full peer review and any attached files.

Reviewer #1: No

Reviewer #2: No

---

## [Author Response · Author response to Decision Letter 1]

7 Oct 2023

1. In the Discussion, the paper has described the treatment of AAI, selection of surgical procedures, and identification. However, the correlation between AAI and body weight, as described in this paper, has no clear pertinence for the treatment and prognosis of AAI.

We modified this sentence as follow.

It is known that some AAI cases do not show postoperative neurological improvement. However, the factors that contribute to this are not known. There are also reports that dorsal decompression should be considered based on preoperative CT and MRI imaging, but no clear guidelines regarding the choice of surgical technique according to comorbidities are recognized [19, 23]. Because this is an observational study, it is unclear whether the correlation between concurrent diseases and body weight in AAI cases has clear implications for the treatment and prognosis of AAI. However, considering these reports and the results of this study, it may possible that weight, detailed preoperative imaging of concurrent diseases and assessment of postoperative neurologic improvement rates may be relevant.

2. Figures 3 and 5 are not labeled in detail, preventing a better understanding of dens dysplasia and intracranial arachnoid cyst.

We modified figures and its figure legends as you mentioned.

---

## [Decision Letter · Decision Letter 2]

11 Oct 2023

Relationship between clinical parameters and malformations in dogs diagnosed with atlanto-axial instability

PONE-D-23-12731R2

Dear Dr. Itoh,

We’re pleased to inform you that your manuscript has been judged scientifically suitable for publication and will be formally accepted for publication once it meets all outstanding technical requirements.

Kind regards,

Miquel Vall-llosera Camps

Staff Editor

PLOS ONE

Reviewers' comments:

Reviewer's Responses to Questions

**Comments to the Author**

1. If the authors have adequately addressed your comments raised in a previous round of review and you feel that this manuscript is now acceptable for publication, you may indicate that here to bypass the “Comments to the Author” section, enter your conflict of interest statement in the “Confidential to Editor” section, and submit your "Accept" recommendation.

Reviewer #2: (No Response)

2. Is the manuscript technically sound, and do the data support the conclusions?

Reviewer #2: (No Response)

3. Has the statistical analysis been performed appropriately and rigorously? 

Reviewer #2: (No Response)

4. Have the authors made all data underlying the findings in their manuscript fully available?

Reviewer #2: (No Response)

5. Is the manuscript presented in an intelligible fashion and written in standard English?

Reviewer #2: (No Response)

6. Review Comments to the Author

Reviewer #2: (No Response)

7. PLOS authors have the option to publish the peer review history of their article (what does this mean?). If published, this will include your full peer review and any attached files.

Reviewer #2: No

---

## [Editor Report · Acceptance letter]

16 Oct 2023

PONE-D-23-12731R2 

Relationship between clinical parameters and malformations in dogs diagnosed with atlanto-axial instability 

Dear Dr. Itoh:

I'm pleased to inform you that your manuscript has been deemed suitable for publication in PLOS ONE. Congratulations! Your manuscript is now with our production department. 

Kind regards, 

on behalf of

Dr. Miquel Vall-llosera Camps 

Staff Editor

PLOS ONE